# Non-Pharmacological Interventions on Pain in Amyotrophic Lateral Sclerosis Patients: A Systematic Review and Meta-Analysis

**DOI:** 10.3390/healthcare12070770

**Published:** 2024-04-01

**Authors:** Marianna Papadopoulou, Apostolos Papapostolou, Rigas Dimakopoulos, Stavroula Salakou, Eleftheria Koropouli, Stella Fanouraki, Eleni Bakola, Christos Moschovos, Georgios Tsivgoulis

**Affiliations:** 1Department of Physiotherapy, University of West Attica, Ag. Spyridonos Str., 12243 Athens, Greece; rdimak@uniwa.gr; 2Second Department of Neurology, Attikon University Hospital, School of Medicine, National and Kapodistrian University of Athens, Rimini 1, 12462 Athens, Greece; apapas55@yahoo.gr (A.P.); bneurologic@attikonhospital.gov.gr (S.S.); ekoropou@med.uoa.gr (E.K.); stelfanou@gmail.com (S.F.); elbakola@yahoo.gr (E.B.); moship@windowslive.com (C.M.); tsivgoulisgiorg@yahoo.gr (G.T.)

**Keywords:** ALS, pain, non pharmacological intervention, muscle exercise, aerobic exercise

## Abstract

Background: Amyotrophic lateral sclerosis (ALS) is a progressive neurodegenerative disorder affecting upper and lower motor neurons. Some ALS patients exhibit concomitant nonmotor signs; thus, ALS is considered a multisystemic disorder. Pain is an important nonmotor symptom. Observational and case–control studies report high frequency of pain in ALS patients and it has been correlated with depression and quality of life. There are no specific scales for the assessment of pain and no randomized controlled trials (RCTs) regarding the drug management of pain in ALS. Aim: To systematically review the evidence for the nonpharmacological interventions (NPIs) in relieving pain in ALS, on March 2024, we searched the following databases: Pubmed, Scopus, Web of Science, and Cochrane. We also checked the bibliographies of trials identified to include further published or unpublished trials. Main results: A total of 1003 records were identified. Finally, five RCTs including 131 patients (64 in the intervention group and 67 in the control group) were included for meta-analysis. The interventions of the included RCTs consisted of muscle exercise, combined aerobics–strength intervention, and osteopathic manual treatment. The meta-analysis did not find a statistically significant difference in favor of NPIs for alleviating pain in ALS patients. Conclusions: ALS has a fulminant course and irreversibly leads to death. Pain in ALS patients, although a common nonmotor symptom, is often unrecognized and undertreated, and this is underlined by the lack of any RCTs on drug therapy for pain. Albeit NPIs are considered safe, as adverse effects are rarely reported, this systematic review did not provide sufficient evidence for a beneficial effect on pain. The scarceness of relevant literature highlights the need for future studies, with larger samples, more homogeneous in terms of interventions and population characteristics (stage of disease), and better choice of measurement scales to further investigate the efficacy, if any, of various pain interventions in ALS patients.

## 1. Introduction

Amyotrophic lateral sclerosis (ALS) is a progressive neurodegenerative disease affecting upper and lower motor neurons, leading to weakness and, eventually, death from neuromuscular respiratory failure. Until recently, ALS was considered a pure motor disease, but involvement of nonmotor areas of the brain is recognized and ALS is now considered a multisystem disorder [1]. Many of these nonmotor symptoms, including cognitive and behavioral impairment [1,2,3], extrapyramidal [3], and autonomic symptoms [1,2,3], are underdiagnosed and left unreported during illness, probably overwhelmed by the fulminant course of the disease.

Although in Awaji, criteria sensory involvement is regarded as an exclusion criteria [4], in the revised El Escorial criteria [5], deficits in sensory system are considered a feature in ALS. Previous studies [6] report sensory involvement in ALS patients, supported by pathologic evidence of sural nerve biopsy, suggesting axonal loss of large myelinated fibers that subserve proprioception and light touch. Involvement of small fibers (Aδ and C), implicated in thermal and pain sensation, has also been demonstrated [7] using a comprehensive quantitative sensory testing (QST) battery. However, alterations in QST were related to thermal detection threshold only and were not associated with the presence of pain.

ALS patients experience pain in a significant proportion, but few studies examine its prevalence. In Hurwitz et al.’s systematic review and meta-analysis [8], pooled prevalence of pain was up to 60% but high heterogeneity was detected between studies. When studies used only validated measures, the prevalence of pain was 65%, but if tailored measures such as interviews, unvalidated questionnaires, or retrospective medical record reviews were used, then pooled prevalence dropped to 45%. They found that pain was most often located in the upper limbs, was of moderate to severe intensity, and was characterized as cramps or spasms. Neuropathic pain was rarely described, in only 3 of the 21 included studies and only in studies that used specific tools to detect neuropathic pain. Chio et al. [9], in their cross-sectional study, found that pain was present in 56.9% of ALS patients, was frequent in all stages of ALS, and its frequency and intensity correlated with a worse functional score. In Lopes et al.’s [7] cross-sectional study, including a sample of 80 ALS patients, chronic pain was reported by 46% of ALS patients, using the visual analogue scale (VAS) as a measurement tool. They found that pain was mostly of musculoskeletal origin, as no patient reported pain with neuropathic characteristics. Pain was mainly located in the head or neck and correlated to anxiety and depression. In Rivera et al.’s cross-sectional study [10] that included 64 ALS patients, pain was reported by 51.6%. Although the Neuropathic Pain Scale was used as a measurement tool, pain was considered to be of musculoskeletal origin. Pain location was equally reported in upper and lower extremities and was described either as dull, sharp, or ache. In Hanisch et al.’s [11] cross-sectional survey, 46 ALS patients were studied. The short-form Brief Pain Inventory (BPI) was used as a measurement tool. Pain prevalence in ALS was up to 78%. The back, the extremities and the joints were most commonly affected. Patients described pain as “miserable”, “tender”, or “dull/pressing”. Regarding intensity, only 33% of patients described pain as moderate to severe. Only 47% of patients received some kind of treatment for pain, either pharmacological or nonpharmacological, although pain interfered with mood and quality of life. In Wallace et al.’s [12] cross-sectional study, including 80 ALS patients, pain was reported by 85% using BPI as a measurement tool. Pain was located more frequently in the limbs and was described as cramping, aching, tiring, sharp, and tender. Pain significantly interfered with everyday activities and mood, thus affecting quality of life.

Pain in most studies is characterized as non-neuropathic, not involving the pain pathways, and is attributed mainly to musculoskeletal problems. Cramps and spasticity, myofascial pain syndrome, and contractures followed by joint tightness [1,13,14] are the main causes of pain in ALS. Other sources of pain might be limb edema, acute injuries from falls, constipation due to immobility and to limited food and fluid intake due of dysphagia, and skin pressure due to limb positioning and decubitus [13].

It is undoubted that prolonged immobility and postural changes are the main causes of pain in ALS and therefore explain and justify the use of nonpharmacological interventions in the treatment of pain. However, pain may be reported by ALS patients early in the course of the disease, even before severe immobility is noted. Thus, immobility, per se, cannot explain all cases of pain that occur in ALS. The mechanism by which pain develops in ALS patients is not clear. Inflammatory injury may cause sensitization not only to nociceptive pathways but to neuropathic ones as well, through damage to peripheral nerves or to the central nervous system [15]. Degeneration of dorsal root ganglia has been hypothesized to explain electrophysiological abnormalities of sensory conduction studies that might contribute to pain perception [16]. Evidence of sensory involvement is further supported by pathologic findings in sural nerve biopsy [6]. Large myelinated fibers were found to be predominantly affected, showing axonal degeneration and regeneration accompanied by myelin abnormalities.

Regardless of the origin and process of its occurrence, pain has a negative impact on the quality of life (QoL) and correlates with depression and anxiety in patients with ALS [7,9,11,17,18]. Despite the high prevalence of pain and its impact on patients, it is rarely reported unless actively sought. Even more, pain like other nonmotor symptoms, such as dyspnea (49%), muscle stiffness (39%), sialorrhea (32%), pseudobulbar affect (29%), and weight loss (22%), pain (44%) is also undertreated since only three quarters of ALS patients with pain are treated pharmacologically [19].

Control of pain is essential in improving QoL and allowing patients to participate in activities of daily life since pain correlates with depression ratings. Nonopioid analgesics are used as first-line drugs followed by opioids [14]. More specifically, for cramps and spasticity, different drugs have been tried, such as baclofen, phenyntoin, gabapentin, benzodiazepine, carbamazepine, tizanidine, dantrolene, leviteracetam, and Cannabis sativa [20,21,22,23]. What is interesting is that there are no RCTs exploring the efficacy of drug therapy for pain in ALS [24,25], only case series, case reports, expert opinions, and caregivers’ experience.

Only a few studies have investigated the effects of nonpharmacological interventions (NPIs) and their role in the care of patients with ALS. Those studies vary depending on the NPI applied. Some studies used aerobic/endurance training [26,27,28,29,30]. Others utilized resistance/strengthening training [31,32,33,34,35], and others a combination of endurance and resistance training [36,37,38,39,40,41,42]. There are also studies on the effects of specific respiratory exercises [43,44]. These studies also vary in the outcome measured. Most of these focus on the effects on muscle strength, cardiovascular and respiratory function, QoL, and increased life expectancy. Recent and past reviews have systematically analyzed the effect of various NPIs. In the Chen et al. review [45], published in 2008, the authors concluded that exercise is more beneficial than deleterious for ALS patients. In Tsitkanou et al.’s review [46], the authors concluded that although there is no evidence of survival prolongation, NPIs of either type of intervention may favorably affect QoL, functionality, muscle strength, and cardiovascular and respiratory function. In Zhu et al.’s review [47], a network meta-analysis of 10 RCTs, the authors found that a combined program of aerobic and resistance exercise showed the greatest improvement on QoL and reduced fatigue. Overall physical function improved after aerobic and resistance training but no clear effect on respiratory function was observed. Due to the paucity of studies in ALS patients, animal studies help to further investigate the effects of NPIs in ALS. Findings suggest that endurance training might be beneficial for ALS SOD1 mice, and the effect depends on the intensity and duration of the exercise as well as the gender. In Veldnik et al.’s study [48], moderate exercise delayed the disease onset and survival was prolonged, but only on female mice. In Kirkinezos et al.’s study [49], treadmill training in SOD1 mice had a beneficial effect on life expectancy on male mice only. In Mahoney et al.’s study [50], training in high-intensity endurance treadmill exercise hastened death by 11 days in male mice. In Liebetanz et al.’s study [51], SOD1 mice had intensive treadmill running. The exercise group showed a favorable survival trend, although not significant.

A multidisciplinary approach has been proposed for the effective management of the complex needs of ALS patients, including pain. ALS places considerable burden not only on patients but on their caregivers as well, who focus not only on treatment strategies but on psychosocial support, which is commonly neglected. A “neuropalliative rehabilitation” model of care, especially through the framework provided by International Classification of Functioning, Disability, and Health (ICF) [52], is suggested to fully embraced the complex needs of ALS patients and their families.

Based on the above, the high prevalence of pain, its complex and diverse presentation and impact on ALS patients, and the paucity of relevant pain management literature, we conducted this study to systematically review the effectiveness of NPI for pain in ALS patients.

## 2. Methods

This systematic review (SR) was conducted according to the PRISMA (Preferred Reporting Items for Systematic Reviews and Meta-Analyses) guidelines to ensure complete reporting [53].

### 2.1. Protocol and Registration

The protocol of this overview can be found on OSF.IO (26 December 2023) 

### 2.2. Literature Search

A comprehensive literature search of randomized controlled trials (RCTs) was performed in March 2024 and included papers published the last 20 years in Pubmed, Scopus, Web of Science, and Cochrane.

The literature search was composed of the Medical Subject Headings (MeSH) and free-text words for (ALS) AND (AMYOTROPHIC LATERAL SCLEROSIS) AND (MOTOR NEURON DISEASE) AND (LOU GEHRIG) OR (PAIN) AND (SMALL FIBER NEUROPATHY) and were implemented for different databases. Different search equations are provided as Appendix A.

Only full text original articles, written in English language, and published in indexed peer-review journals were eligible for inclusion. Cross-sectional studies, case reports, published abstracts, dissertation materials, and conference presentations were not included.

The reference lists of all the appraised articles were screened for relevant citations that might have been missed from the electronic searches. Once all articles were found, duplicate articles were removed. Two reviewers (M.P. and A.P.) with long clinical and academic experience in the diagnosis and treatment of patients with ALS independently screened the titles and abstracts for eligibility and examined the full text of the articles for final decision [54] (Figure 1). 

### 2.3. Inclusion Criteria

Only RCTs were included.Studies including patients with a diagnosis of ALS without restriction in terms of age, sex, and stage and duration of the disease.Any NPI with a structured protocol, including physical exercise.Evaluation of pain as outcome, either primary or secondary, regardless of the measurement tool used.

### 2.4. Exclusion Criteria

ALS was not the main disease, or patients had also other neuromuscular dysfunction not related to ALS.Concurrent diseases that may lead to pain, e.g., rheumatologic disease.Pharmacological intervention was the only intervention without information on NPIs.Animal studies, reviews, case reports, or case series studies.

Individual searches were compared and in case of disagreement, a consensus was made regarding the inclusion of each study. If needed, a third reviewer was consulted (R.D.).

### 2.5. Data Extraction

Both reviewers extracted data from 5 eligible studies, which included authors, date of publication, number of subjects, epidemiologic data (age, sex), disease duration and severity, NPIs (type and protocol design), tool of outcome measure (pain), and main findings. The characteristics and results of the included RCTs are presented in Table 1.

### 2.6. Risk of Bias

Two independent authors (M.P. and R.D.) evaluated the quality of the selected studies using Cochrane risk of bias tool, the RoB 2 tool [55]. The tool comprises of the following domains: randomization process, deviations from the intended interventions, missing outcome data, measurement of the outcome and selection of the reported result. According to the Cochrane Collaboration Risk of Bias Assessment Tool, a study is judged to be at low risk of bias if it is of low risk for all domains and is judged to raise some concerns if at least one domain raises some concerns, but not to be at high risk of bias for any domain. Finally, a study is considered of high risk if it is judged to be at high risk of bias in at least one domain.

### 2.7. Data Synthesis and Statistical Meta-Analysis

To analyze the effect of NPIs on pain and in ALS patients, we estimated the weighted mean differences and 95% confidence intervals (Cis) from each study using MetaView Review Manager Version 5.4 [56]. For weighted mean differences, a point estimate of zero reflected “no effect”, and less than zero favored the NPI intervention. Statistical heterogeneity was assessed by using the χ2 test (*p* < 0.1). The I^2^ statistic was also calculated, and we considered I^2^ > 50% to indicate significant heterogeneity across studies [57].

**Table 1 healthcare-12-00770-t001:** Characteristics of the included studies.

Study	Sample Size (IG/CG)	AGE (Mean-SD) (IG/CG)	SEX (F%) (IG/CG)	Disease Duration Months (Mean-SD) (IG/CG)	Disease Severity ALSFRS(-R) (Mean-SD) (IG/CG)	Intervention Duration	Control	Outcome Measure (Primary/Secondary)	Findings
Dal Bello Haas et al., 2007 [58]	8/10	56.0 (7.3) /51.8 (12.6)	4/7 31%/50%	20.4 (12.8)/15.4 (13.0)	35.2 (3.6)/33.6 (2.4)	6M home exercise. 3/w U/E and L/E moderate-load and moderate Intensity resistance exercise program.	Once-daily U/E and L/E stretching exercises.	SF36 Bodily pain subscale SECONDARY	Pain scores did not differ between groups.
Drory et al., 2001 [32]	8/6	58.0 (13.2) /60.7 (16.4)	6/5 44%/36%	20.7 /19.4	27.5	Individual muscle exercise 15 min 2/d at home: 3, 6 (9 and 12) months.	No exercise apart usual day life requirements.	VAS PRIMARY	In both groups pain increased over time. Exercise had no obvious effect.
Karlon et al., 2021 [37]	14/14	58.5 (13.2) /60.4 (14.7)	11/6 78%/42%	7.3 (12) /6.4 (12.2)	35.7 (5.3) /37.5 (5.6)	24 sessions (2/w): aerobic training 20–30 min flexibility 10 min strength training 20 min 3 M.	60 session (5/w) basic stretching exercises 20 min.	SF 36 SUBSCALE SECONDARY	No differences were found in the SF 36 categories.
Maggiani et al., 2016 [59]	7/7	54.0 (11.6) /51.0 (6.5)	2/2 28%/28%	17.1 (4.5) /38.0 (30.5)	35.0 (6.7) /40.4 (4.0)	Osteopathic manual treatment 1/w for 4 w ½ w for 8 w (open) 4 W.	Physiotherapy (2/w).	BPI SECONDARY	No BPI differences were found.
Van Groenestijn et al., 2019 [36]	27/30	60.9 (10.0) /62.1 (10.7)	9/8 33%/27%	15.5 (10.9) /18.0 (14.0)	42.3 (3.5) /42.3 (4.2)	2/w home-based training on a cycle ergometer 25–30 min. 1/w supervised individual training sessions 60 min 6 M.	Neuropalliative care by multidisciplinary, secondary care teams.	VAS PAIN SECONDARY	No significant difference.

## 3. Results

### 3.1. Characteristics of Included Trials

Five RCTs were included in this systematic review (Figure 1). Descriptive characteristics and results of all included studies are presented in Table 1. In total, 131 ALS patients were included (64 in the intervention group, 67 in the control group). The number of participants in each study varied from 14 to 57. The dropout rate was high in some studies [32,36,58] mainly due to disease progression and death in some cases.

Figure 2 shows the assessment results of the risk of bias of the included studies. The randomization process was appropriate for all studies. Inevitably, participants were aware of their assigned intervention during the trial, as were carers and people delivering the interventions, because of the nature of the intervention. Another limitation in these studies was the high rate of dropouts, due to the nature of the disease and its rapid progression that could have influenced the outcome’s true value. The method of measuring the outcome was appropriate in all studies, but in one study [32] it was not clearly stated whether outcome assessors were blind to the intervention received by study participants. Regarding the last domain, there were no concerns of selecting reported results, other than the prespecified intended.

### 3.2. Characteristics of Participants

Average age ranged from 51 to 62.1 years. All studies included patients of both sexes. Regarding disease characteristics, all studies included patients in mild to moderate disease stage, as measured by ALS Functional Rating Scale (ALSFRS) or its revised form, and disease duration ranged from 7 months (the least) [37] to 38 months (the most) [59].

### 3.3. Characteristics of Interventions

Interventions across studies varied. Stretching and resistance exercise was administered by Dal Bello-Haas’s study [58] for upper and lower limbs. The exercise was performed daily and was of moderate load and of moderate intensity. If a muscle group was too weak, with a muscle strength grade less than 3, this muscle group was omitted from the exercise program. Evaluation was performed once a month and consisted not only of muscle strength but of respiratory function as well to monitor for adverse effects. In Drory et al.’s [32] study, patients also received an exercise program involving most muscle groups of the four limbs and trunk in order to improve muscle endurance, having the muscles work against only modest loads but undergo significant changes in length. The exercise program lasted 15 min and was performed twice daily at home. Patients were contacted by telephone every 14 days to check compliance. In Kalron et al. [37] and in the van Groenestijn study [36], patients received a combined aerobic and strength program. Aerobic training was administered twice a week in both studies and consisted of recumbent cycling [37] 20–30 min per session or training on a cycle ergometer, step board, and treadmill [36], 20–35 min per session. The exercise program consisted of stretching and functional exercises [37], 30 min per session, focusing on the large muscle groups of the trunk and upper and lower limbs, or 20 min of strengthening exercises of the quadriceps, biceps, and triceps muscles [36]. Finally, Maggiani et al. [59] applied osteopathic manual treatment (OMT) that was administered once a week for the first four weeks, and fortnightly for the following eight weeks. Treatment duration was always 40 min. The control group in all studies received usual care, which varied from instruction not to perform any physical activity other than their usual daily life requirements [32], to performing daily upper and lower limb stretching exercises [37,58] or standard care treatment with multidisciplinary care teams [36,59]. The duration of intervention also varied between studies, ranging from 4 weeks [59] up to 3 months [37] and up to 6 months in the remaining studies [32,36,58].

The intervention program was tailored to each patient’s tolerance and limitations, including general health status and actual fitness level [32,58,59], and heart rate and oxygen saturation levels were monitored throughout to ensure safety of patients [36,37] such that exercise intensity did not result in an increase in heart rate greater than 75%. All adverse events such as fatigue, cramps, myalgia, increase in fasciculations, restless legs, and injuries were systematically recorded. No serious side effects affecting the safety of the intervention or leading to treatment discontinuation were reported.

### 3.4. Characteristics of Outcomes Measured

Pain was the primary outcome in only one study [32] and a secondary in the other four studies [36,37,58,59]. VAS was used by two studies for pain measurement [32,36], Short Form-36 (SF-36) subscale by two studies [37,58] and one used BPI short-form BPI (59). VAS evaluates pain using a straight line with one end meaning no pain and the other end meaning the worst pain imaginable. BPI measures pain on a 11-point scale (from 0–10), where higher scores indicate worse pain and interference, and item 22 in SF-36 rates from 1–5 how much pain interfered with the patient’s normal work during the past 4 weeks.

In addition to pain, all included studies measured QoL, either using SF-36 [32,36,37,58] or the McGill QoL (16 items) [59]. In one study [36], a second scale, the disease-specific health-related quality of life (HRQoL) was also used. Disease progression was also measured by most studies by monitoring changes in ALS Function Rating Scale [32,36,37,58]. Fatigue was the next most common measured outcome, either by Fatigue Severity Scale (FSS) [32,37,58] or by the Checklist Individual Strength subscale fatigue (CIS-fatigue) [36]. Other parameters measured were respiratory function [36,37,58], muscle strength [32,36], mobility [36,37], spasticity [32], sleep [36], and perceived satisfaction regarding the treatment alongside with the recording of adverse effects [59].

### 3.5. Effectiveness of NPI on Pain in ALS Patients

Meta-analysis showed no statistically significant differences between the intervention and control groups, regardless of the type or duration of the intervention. The forest plot describing data synthesis was generated using MetaView Review Manager Version 5.4 [56] (standardized mean difference (SMD): 0.10; 95% confidence interval: −0.24 to 0.45; Z = 0.60; *p* = 0.55 showing no heterogeneity, chi^2^ = 1.14, I^2^ = 0%) (Figure 3).

To further explore the effect of various NPIs, we ran a sensitivity analysis, by separately studying the included RCTs on the base of the applied intervention, resistance exercise, combined exercise (aerobic–resistance), and OMT. The results demonstrated that resistance exercise had no positive effect size (SMD: 0.01; 95% confidence interval: −0.69 to 0.71; Z = 0.03; *p* = 0.98 showing no heterogeneity, chi^2^ = 0.03, I^2^ = 0%), nor did combined exercise (SMD: 0.14; 95% confidence interval: −0.30 to 0.57; Z = 0.62; *p* = 0.54 showing minimal heterogeneity, chi^2^ = 1.01, I^2^ = 1%) and OMT [59] (SMD: 0.14; 95% confidence interval: −0.91 to 0.50; Z = 0.26; *p* = 0.79 showing minimal heterogeneity, not applicable (Figure 3).

Regarding the outcome measures other than pain, in some cases, some difference was observed. The exercise training administered by two studies [32,58] resulted in less decline in global functioning, as measured by ALSFRS. In the Bello-Haas et al. study [58], the resistance exercise group showed significantly smaller decline in the SF-36 P-F subscale score at 6 months compared with the control group. In the study by Drory et al. [32], there was a positive effect on spasticity in ALS patients receiving the exercise training, but this positive effect was retained up to three months and was not maintained at six months. In the Kalron et al. study [37], which used a combined program of aerobic and resistance training program, global functioning, mobility, and respiratory function were also preserved in the investigational group compared with the control group, but no other significant difference was found. In the Maggiani et al. study [59] that administered OΜΤ, safety and feasibility of treatment, which were the primary outcomes, achieved satisfactory results. Finally, in the study by van Groenestijn et al. [36], who also used a combined program of aerobic and strengthening exercises, no outcome measure, either primary or secondary, showed any statistically significant difference compared to the usual care group.

## 4. Discussion

This systematic review examined the effects of different NPIs on patients suffering from ALS across five RCTs. In total, 131 patients were included. The meta-analysis showed that, overall, NPIs, no matter which, did not significantly reduce pain or favorably affect other outcome measures, with minor exceptions.

Two previous reviews investigated the effectiveness of NPIs on various outcomes in ALS patients, pain being one of them, and reached the same conclusion. In their review, Rahmati and Malakoutinia [60], on the effects of aerobic, resistance, and combined exercise training for patients with ALS, failed to demonstrate any statistically significant difference in any outcome except Vo2peak and overall QoL. When low-quality studies were omitted, even this effect disappeared. The second review [33], published more than 10 years ago, also investigated the effect of therapeutic exercise in ALS patients for a long list of outcomes. This review included only two RCTs and reached the same conclusion, that only functional score differed between groups, and not pain, or any other measured outcome.

The present review differs from the previous, since it is focused only on pain as an outcome measure and, in addition, the current meta-analysis includes further studies and therefore extends and validate previous results.

Apart from the abovementioned reviews on the effects of NPIs on pain in ALS patients, other reviews have investigated the effect of NPIs on other outcomes. Zhu et al. (2022) [47], in their network meta-analysis, studied the effect of different exercise interventions across 10 RCTs on overall functional score, respiratory function, perceived fatigue, and quality of life. They concluded that a combined program of aerobic and resistance exercise might be the best to improve QoL and reduce fatigue. The overall functional score was improved by either aerobic or resistance interventions. No clear conclusion emerged regarding the effect on respiratory function. Chen et al. (2008) [45] reviewed the role of exercise in ALS patients and investigated its benefits and how they are measured. Regarding resistance exercise, they reported a case study [31] where resistance exercise had beneficial effect on muscle strength in most muscle groups, resulting in a subjective functional improvement. They also discussed the results of two studies included in our review, the Bello-Haas and Drory studies [32,58], focusing mainly on functional improvement as measured by the ALSFRS, which showed improvement at 6 and 3 months, respectively. Regarding aerobic exercise, they reported that those studies [29,30,61,62] focus mainly on metabolic parameters, such as oxygen cost, lipid metabolism, lactate values, and precocious anaerobic threshold, and less on functional outcomes and QoL. Finally, Tsitkanou et al. (2019) [46] reviewed 10 preclinical studies on the role of exercise training on SOD1 mice, a mouse model of familial ALS mice, and 12 clinical studies in ALS patients. They concluded that despite mild-to-moderate exercise seeming to improve the survival of ALS mice, this is not supported in clinical studies, although there is a positive effect on QoL, functionality, muscle strength, and cardiorespiratory function. They attributed this discrepancy to the limitations of clinical studies in ALS patients, namely, small sample size, inability to control for confounding factors, heterogeneity of the disease, and difficulty in creating a representative control group for RCTs.

NPIs are emerging as an alternative therapeutic intervention, in various medical conditions, as a complementary medical treatment or as a monotherapy. NPIs show significant benefits compared to medication; they do not interact with other drugs given concomitantly, do not require good renal and liver function, do not affect laboratory tests, and, in general, lack side effects, apart from increased soreness or muscle pain. On the other hand, NPIs are time-consuming and require active participation of the patient in the process, and sometimes it is necessary for the patient to visit a special rehabilitation center.

Previous systematic reviews [63,64] and meta-analyses on the effect of NPIs on chronic pain in musculoskeletal conditions showed favorable results. They found that exercise reduces pain severity and might have variable effects on QoL, with few adverse effects. Yet, the exact mechanism via which exercise reduces pain is still unclear. Factors such as self-efficacy, conditioned pain modulation, and central nervous system adaptation in response to an exercise program may contribute to pain reduction. NPIs have also been proven effective in cases of neuropathic pain, as in cases of chemotherapy-induced polyneuropathy (CIPN) [65]. Possible mechanisms that have been proposed include modulation of nociception and increase in pain threshold.

Based on the above, NPIs are shown to be effective and safe in various pain conditions. On the contrary, in the case of ALS, exercise programs, either aerobic, resistance, or combined, or any other kind of NPIs, such as OMT, did not prove to be effective for pain. Several factors might be responsible for this negative result. The relative literature is quite poor. RCTs are rare, of low to moderate methodological quality, and therefore at high risk of bias, and mainly consist of small sample sizes. The dropout rate in included studies is high, affecting the results. Moreover, there is not a specific-to-ALS pain scale, and each study used either a general pain scale or a subscale derived from the QoL questionnaire (SF-36]. The lack of a specific measurement tool may affect the sensitivity with which pain and its changes are recorded.

Another limitation in studying pain in ALS patients is the inevitable selection of patients who are in the early stages of the disease in order to be able to follow the exercise program. But, because the disease progresses rapidly, patients may be moved to a more severe stage and may be unable to complete the planned program. Due to the progression of the disease, respiratory function should be monitored throughout the intervention to avoid adverse effects.

Another important factor might be the nature of the disease. ALS has a fulminant course. Patients are aware of the rapid and malignant course of their disease, leading inevitably to death. This critical psychological state can act as an obstacle to any therapeutic intervention that does not aim at a cure. Probably, this is the reason why pain is underreported spontaneously by patients, unless they are asked directly. Exercise failed to improve QoL, probably due to the reasons stated above. The same has been noticed in other critical diseases, as in CIPN, where, although NPIs showed favorable outcome in pain, they failed to do so regarding QoL [65]. In this case, it was suggested that pain relief alone is not sufficient to improve QoL when patients have to face the prospect of a chronic disease with a not-always-favorable outcome.

Interestingly, there are no RCTs on the pharmacological treatment of pain in ALS [25], and, therefore, there are no definitive conclusions on their effectiveness. As a result, pain management is primarily guided by small case series, case reports, expert opinion, and caregiver experience.

## 5. Conclusions

NPIs, although effective in other medical conditions, were not found to be effective for pain in ALS patients and showed a high dropout rate. Furthermore, only a small percentage of patients, who are in early stages, can be recruited, but even then, they should be closely monitored as disease progression and, more specifically, respiratory deterioration, may make them unsuitable for exercise programs.

Based on the above, NPIs, although generally considered safe, should be used with caution in patients with ALS. More RCTs, with larger sample sizes and higher methodological quality, are needed to further investigate their efficacy and patient tolerability.

## Figures and Tables

**Figure 1 healthcare-12-00770-f001:**
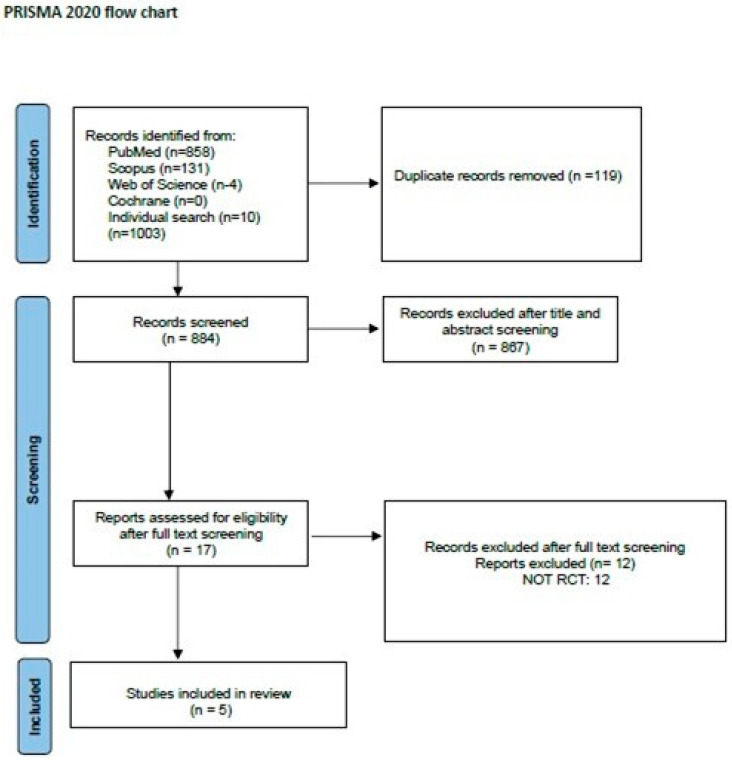
PRISMA flow chart for the study selection process [54].

**Figure 2 healthcare-12-00770-f002:**
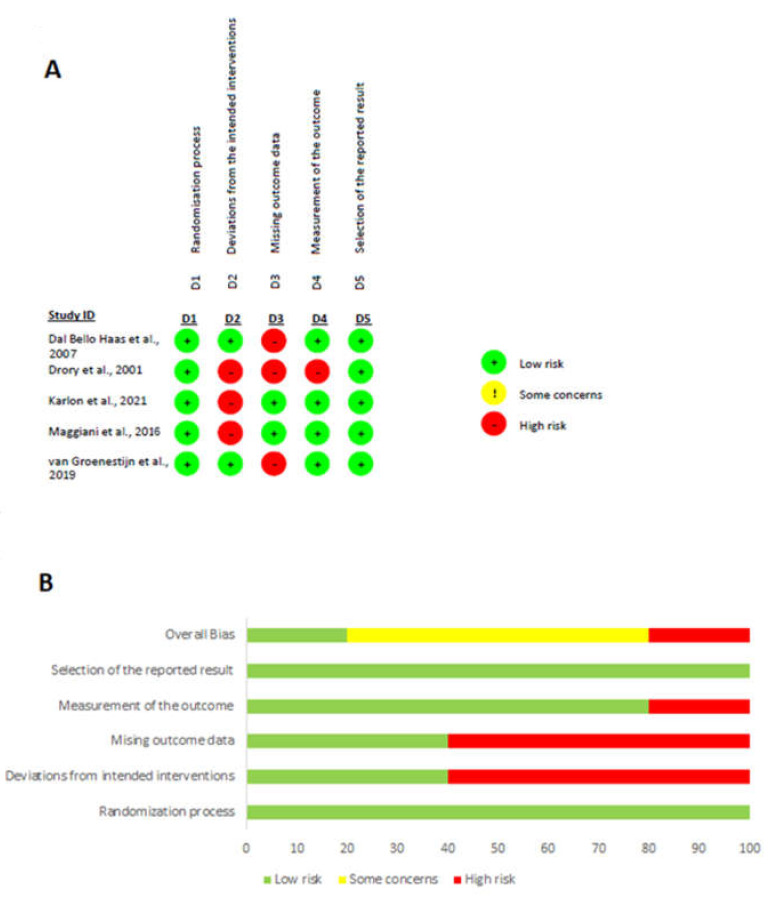
The result of the risk of bias assessment. (**A**) Risk of bias graph; (**B**) risk of bias summary [32,36,37,58,59].

**Figure 3 healthcare-12-00770-f003:**
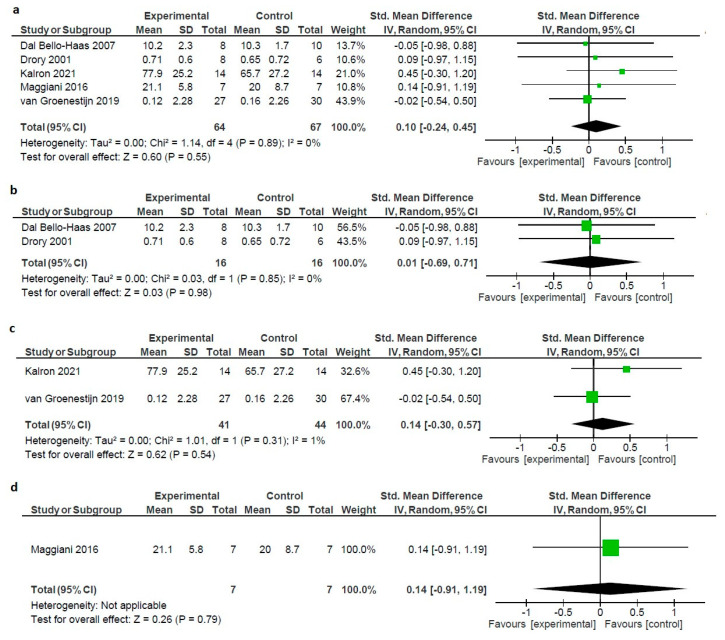
(**a**) Forest plot of the effect of all types of interventions on pain in ALS patients. (**b**) Forest plot of the effect of exercise on pain in ALS patients. (**c**) Forest plot of the effect of combined on pain in ALS patients. (**d**) Forest plot of the effect of OMT on pain in ALS patients [32,36,37,58,59].

## Data Availability

No new Data was created.

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
