# Peer review of "Non-Pharmacological Interventions on Pain in Amyotrophic Lateral Sclerosis Patients: A Systematic Review and Meta-Analysis"

_healthcare, 2024, doi:10.3390/healthcare12070770_

Round 1
Reviewer 1 Report
Comments and Suggestions for Authors
We have proceeded to analyse each of the sections of the manuscript entitled: Non-Pharmacological Interventions on Pain in Als Patients. A 2 Systematic Review and Meta-Analysis. Below are the most relevant aspects that need to be addressed in order to improve the methodological quality of the manuscript.
Introduction:
The overall number of references throughout the manuscript is extremely low for a review article. For example, there are important reviews on this topic that are not included such as: Tsitkanou, S., Della Gatta, P., Foletta, V., & Russell, A. (2019). The role of exercise as a non-pharmacological therapeutic approach for amyotrophic lateral sclerosis: beneficial or detrimental?. Frontiers in neurology, 10, 783. Rugo expand references and background construct on this topic in this section.
Material and methods:
I would like to recommend the use of a table with the different search equations and include this as supplementary material.
On the other hand the use of only two databases is insufficient and I recommend to extend the search to at least two more databases.
Figure one has very poor image quality.
We recommend incorporating the Rob2 tool for risk of bias analysis.
Results:
Figure two because a random effects model is used rather than a fixed effects model as heterogeneity is low?
Discussion
Too short on references and does not discuss the results with other previous reviews on the same topic.
Thank you for giving me the opportunity to review this paper. I hope the comments will improve the quality of the manuscript.
Author Response
RESPONSE TO REVIEWER #1
R: The overall number of references throughout the manuscript is extremely low for a review article.
A: We would like to thank the Reviewer for this critical remark. We have made every effort through our systematic literature review to include all studies investigating the effect of NPIs on pain in ALS patients.
But, as we have stated in Introduction lines 57-58 “ALS patients experience pain in a significant proportion, but few studies examine its prevalence”, and even less is the literature on pain management, lines128-129: “Based on the above, the high prevalence of pain, its complex and diverse presentation and impact on ALS patients, and the paucity of relevant pain management literature”. Furthermore, only two Systematic Reviews have previously “investigated the effectiveness of NPIs on various outcomes in ALS patients, pain being one of them (lines 315-316)”.
There are several studies and important reviews investigating the effect of NPIs on various aspects of ALS patients, but not on pain. The proposed review by Tsitkanou et al is one of them, and we are very happy to include it in our paper.
To further expand the references and background structure on the effect of NPIs in ALS on outcomes other than pain, we have added a new paragraph in the Introduction Section.
The reference list is now expanded from 40 to 64.
- I would like to recommend the use of a table with the different search equations and include this as supplementary material. On the other hand the use of only two databases is insufficient and I recommend to extend the search to at least two more databases.
- We would like to thank the reviewer for this critical observation. We have used COCHRANE and Web of Science in our query, but retrieved very few results. We have now updated the search, added the results of these two databases, and the search equations are provided as supplementary material.
- Figure one has very poor image quality.
- Figure 1 has been updated and image quality was improved.
- We recommend incorporating the Rob2 tool for risk of bias analysis.
- Following reviewer’s recommendation, we have incorporated the Rob2 tool for risk of bias analysis.
- Results: Figure two because a random effects model is used rather than a fixed effects model as heterogeneity is low?
- “Which model to use depends on the circumstances. Generally, the random-effects model is often the appropriate model, capturing uncertainty resulting from heterogeneity among studies. For example, the effect size might be higher or lower in trials where the participants’ demographics vary (eg, older vs younger, or lower vs higher socioeconomic status, or less vs higher educated), or when a different surgical technique of an intervention is used.” Dettori JR, Norvell DC, Chapman JR. Fixed-Effect vs Random-Effects Models for Meta-Analysis: 3 Points to Consider. Global Spine J. 2022 Sep;12(7):1624-1626. doi: 10.1177/21925682221110527. Epub 2022 Jun 20. PMID: 35723546; PMCID: PMC9393987.
In our case, interventions varied between studies, as did participant demographics (age, disease duration and severity). For this reason, we chose the random effects model. After running this model, heterogeneity was found to be very low. We additionally run the fixed effects model which yielded similar results. We chose to retain the original analysis, the random-effects model, as this was our original approach, and since the results did not change when switched to the fixed-effects model.
- Discussion Too short on references and does not discuss the results with other previous reviews on the same topic.
- Οn the topic of our review, namely the efficacy of NPIs on pain in ALS patients, only two reviews have been published, one by Rahmati and Malakoutinia (2021) and the other by Dal Bello-Haas and Florence (2013), that are already discussed in our paper.
We have identified 3 reviews, Zhu et al (2022), Chen et al (2008) and Tsitkanou et al (2019) that study the effect of NPIs in ALS patients on outcomes other than pain, such as overall functional score, respiratory function, oxidative stress, autonomic response (heart rate and ventilatory response), perceived fatigue and quality of life. Following reviewer’s suggestion, all three reviews have been incorporated in the discussion section, and their results on those other outcomes are discussed.
Thus, the reference list has been expanded from 40 to 64.
Once again, we would like to thank Reviewer #1 for her/his insightful and constructive comments on the text, which shed light on some deficiencies that needed to be addressed. We believe this has helped us improve the manuscript substantially.
Reviewer 2 Report
Comments and Suggestions for Authors
Peer review report: Non-Pharmacological Interventions on Pain in Als Patients. A Systematic Review and Meta-Analysis
In this review, the authors systematically reported the results of randomized controlled trials on non-pharmacological intervention for pain in amyotrophic lateral sclerosis (ALS) patients. I believe the topic is of interest to the journal readers. The manuscript is well-written, organized, and thorough. I can only provide minor comments below.
1. The flow chart indicated a large number of records being excluded, 715 out of 733, through two independent reviewers. I suggested the authors include the reviewers’ expertise and perspectives in this area of research/practice.
2. I suggest spelling out each word of the term amyotrophic lateral sclerosis in the manuscript’s title rather than using an abbreviation.
3. I suggest consistently using capital/small letters in abbreviations throughout the manuscript. For example, the term quality of life should be expressed as either QoL or QOL, not interchangeably.
4. I assume Healthcare will perform a spelling check for minor errors.
I want to applaud the authors for doing the job well.
Author Response
RESPONSE TO REVIEWER #2
- In this review, the authors systematically reported the results of randomized controlled trials on non-pharmacological intervention for pain in amyotrophic lateral sclerosis (ALS) patients. I believe the topic is of interest to the journal readers. The manuscript is well-written, organized, and thorough.
- We would like to thank Reviewer #2 for reviewing our manuscript and her/his valuable feedback
- R. The flow chart indicated a large number of records being excluded, 715 out of 733, through two independent reviewers. I suggested the authors include the reviewers’ expertise and perspectives in this area of research/practice.
- Reviewer 1, M.P., is a Neurologist, Associate Professor at the University of West Attica, specializing in Clinical Neurophysiology, working in the diagnosis of ALS patients in a tertiary referral University Hospital. In addition, she has been involved in the implementation of the International Classification of Functioning (ICF) at the national level for the assessment of patients with disabilities. Reviewer 2, A.P., is a Neurologist, MD, PhD, specializing in Clinical Neurophysiology, working in the outpatient diagnosis of ALS patients. Both reviewers, in addition to long clinical experience, have authored scientific papers on ALS and clinical neurophysiology.
The above information, has been added in summary in the text (Methods; Literature Search).
- R. I suggest spelling out each word of the term amyotrophic lateral sclerosisin the manuscript’s title rather than using an abbreviation.
- We would like to thank Reviewer #2 for this thoughtful suggestion. The Title has been changed accordingly.
- R. I suggest consistently using capital/small letters in abbreviations throughout the manuscript. For example, the term quality of lifeshould be expressed as either QoL or QOL, not interchangeably.
- We would like to thank Reviewer #2 for this critical remark. The term “quality of life” was corrected to QoL throughout the manuscript.
- R. I want to applaud the authors for doing the job well.
- Once again, we would like to thank reviewer #2 for reviewing our manuscript and her/his valuable feedback.
Reviewer 3 Report
Comments and Suggestions for Authors
Dear authors, I enjoyed reading the paper entitled " NON-PHARMACOLOGICAL INTERVENTIONS ON PAIN IN ALS PATIENTS.
SYSTEMATIC REVIEW AND META-ANALYSIS." submitted by you to the journal.
As you underlined, the topic covered is fascinating, even if little known.
The paper's writing is excellent, one of the few times in which it is difficult to find aspects to review.
Reviewing each paragraph, I can comment:
Abstract: to reduce it and make it more quickly readable, I suggest removing the inclusion and exclusion criteria that you clearly defined in the body of the text.
Introduction: Excellent. It systematically summarizes the two important aspects to introduce—ALS for pain therapists who are not familiar with this pathology and the pathophysiology of pain for physiatrists and neurologists who could underestimate this symptom.
Methods: excellent presentation and description
Results: precise and exhaustive
Discussion: I would eliminate or summarize the point on line 338, mentioning CIPN and other non-related pains (proper to mention them for hypotheses but not so necessary in its extended discussion.
Apart from these tiny things, I congratulate you again for the methodological quality of the submitted paper.
Author Response
RESPONSE TO REVIEWER #3
Dear authors, I enjoyed reading the paper entitled "NON-PHARMACOLOGICAL INTERVENTIONS ON PAIN IN ALS PATIENTS. SYSTEMATIC REVIEW AND META-ANALYSIS." The paper's writing is excellent, one of the few times in which it is difficult to find aspects to review.
We would like to thank Reviewer #3 for reviewing our manuscript and her/his valuable feedback.
Abstract: to reduce it and make it more quickly readable, I suggest removing the inclusion and exclusion criteria that you clearly defined in the body of the text.
We would like to thank reviewer #3 for this thoughtful suggestion. Inclusion and exclusion criteria were removed from the Abstract.
Introduction: Excellent. Methods: excellent presentation and description. Results: precise and exhaustive
Once again, we would like to thank reviewer #3 for reviewing our manuscript and her/his valuable feedback.
I would eliminate or summarize the point on line 338, mentioning CIPN and other non-related pains (proper to mention them for hypotheses but not so necessary in its extended discussion.t
We would like to thank reviewer #3 for this critical remark. We have summarized the report in CIPN as suggested and removed the extended discussion.
Round 2
Reviewer 1 Report
Comments and Suggestions for Authors
The authors have integrated the proposed improvements, so it can be published in the current version as far as I am concerned.